# Non-Deep Active Learning for Deep Neural Networks

**DOI:** 10.3390/s22145244

**Published:** 2022-07-13

**Authors:** Yasufumi Kawano, Yoshiki Nota, Rinpei Mochizuki, Yoshimitsu Aoki

**Affiliations:** 1Department of Electronics and Electrical Engineering, Faculty of Science and Technology, Keio University, 3-14-1, Hiyoshi, Kohoku-ku, Yokohama 223-8522, Japan; ykawano@aoki-medialab.jp; 2Meidensha Corporation, 2-1-1, Osaki, Shinagawa, Tokyo 141-0032, Japan; nota-y@mb.meidensha.co.jp (Y.N.); mochizuki-r@mb.meidensha.co.jp (R.M.)

**Keywords:** active learning, annotation, uncertainty sample selection

## Abstract

One way to improve annotation efficiency is active learning. The goal of active learning is to select images from many unlabeled images, where labeling will improve the accuracy of the machine learning model the most. To select the most informative unlabeled images, conventional methods use deep neural networks with a large number of computation nodes and long computation time, but we propose a non-deep neural network method that does not require any additional training for unlabeled image selection. The proposed method trains a task model on labeled images, and then the model predicts unlabeled images. Based on this prediction, an uncertainty indicator is generated for each unlabeled image. Images with a high uncertainty index are considered to have a high information content, and are selected for annotation. Our proposed method is based on a very simple and powerful idea: select samples near the decision boundary of the model. Experimental results on multiple datasets show that the proposed method achieves higher accuracy than conventional active learning methods on multiple tasks and up to 14 times faster execution time from $1.2\;\times \;10^{6}$ s to $8.3\;\times \;10^{4}$ s. The proposed method outperforms the current SoTA method by 1% accuracy on CIFAR-10.

## 1. Introduction

In recent years, deep learning models have achieved great success in many fields, but their training relies on large-scale labeled data. However, the annotation of large datasets is difficult, time-consuming, and costly, which is a major challenge for deep learning. In order to reduce the cost of annotation, unsupervised learning [1,2,3], semi-supervised learning [4,5], and active learning [6] have attracted attention.

Unsupervised learning, which learns only from unlabeled images, and semi-supervised learning, which learns from a small number of labeled images and a large number of unlabeled images, aim to train a model by fully utilizing unlabeled images. Rather than annotating all of the data when creating a labeled dataset, active learning aims to reduce the total amount of annotation required by prioritizing images that would be the most effective in training a model.

As shown in Figure 1, active learning selects the most useful images from the unlabeled images for model learning, then labels the selected images using Oracle (annotater), and finally adds the labeled images to the labeled pool to update the task model for learning. This process is repeated until the performance of the task model meets the requirements or the budget is exhausted. Active learning is widely used in image classification [7,8,9], which is the task of categorizing objects in an image, and segmentation [10,11], which is the task of classifying each pixel in an image according to the object to which that pixel belongs.

In this study, we propose a method to select effective images from an unlabeled pool that are useful for training task models in an active learning framework. To determine the efficient images, our method uses an uncertainty sampling approach. The uncertainty sampling approach annotates the data with the highest level of uncertainty in the results when inferring the model. In recent years, there have been many conventional methods using the uncertainty sampling approach. Refs. [12,13,14,15,16] use variational autoencoder (VAE) and Ref. [17] uses deep learning models to infer image loss to select uncertain images. In these methods using a VAE, VAE reduces an image to a low-dimensional representation, and the discriminator determines whether the obtained representation was derived from a labeled or unlabeled image. In the learning-loss method, their model consists of a task module and a loss prediction module that predicts the loss of the task module. These two modules are learned simultaneously, and the loss prediction module is trained to estimate the target loss for unlabeled samples based on mid-layer feature information as a proxy for model uncertainty.

These prior studies rely on specialized learning methods such as adversarial learning. This tends to result in complex and inefficient pipelines that are difficult to use in practical applications. Specifically, they have a model for computing uncertainty in addition to the task model, and since these are independent of each other, it is difficult for the task model to select the images it really needs, and the number of computation nodes is large, which increases the computation time. The disadvantage of the time required is that it detracts from the objective of improving the efficiency of annotation.

To solve this problem, Refs. [18,19,20] have been proposed to compute uncertainty algorithmically, using only task model learning. These methods do not require learning to predict uncertainty, making them easy to use in practice. We propose an uncertainty indicator generator (UIG), which is an algorithmic method similar to these methods. The UIG is a non-deep learning model based on a very simple and powerful idea: selecting samples near the decision boundary of the model. It is designed to express the uncertainty of each image in the unlabeled pool with a specific importance. It calculates the uncertainty indicators based on the prediction vectors of the task model, which has the advantage of allowing the task model to choose the images it really needs. For example, in the case of image classification, the prediction is a probability vector for each category. Images are measured in terms of uncertainty from the two algorithms similar to least confident [6] and margin sampling [6]. In Figure 2, the samples selected by SRAAL [13] and the proposed method are marked with black faction marks. It shows that the proposed method is actually able to select samples near the decision boundary (see Section 4 and Section 5 for details).

Since the proposed method does not require additional training to compute uncertainty, it can update the labeled set at shorter intervals than conventional methods when actually annotating. In addition, the simplicity of the method makes it easy to incorporate into the training pipeline of a task model, and the method does not require hyperparameters, making it easy to apply in real-world applications. UIG can be implemented in a few lines of code. Therefore, images to be annotated can be selected with a few lines of code added to the task model training pipeline. In addition, the UIG does not need to be trained at the dataset creation site, and the task model can be updated at a high frequency because image selection can be performed from scratch faster than with conventional methods.

The main contributions of this paper are the following:The method can be applied to multiple tasks;Experimental results show that it is able to select more uncertain (high loss) images than conventional methods;It is 14 times faster in execution time compared to conventional methods.

## 2. Related Works

There are three main scenarios for active learning [6]: The first is membership query synthesis, which generates valid data for model training. The second is stream-based selective sampling, which sifts the data to label or discard. The third is pool-based sampling, which selects the most effective images for model learning from an unlabeled pool of data. Of the three, pool-based sampling, shown in Figure 1, is the most common [12,13,15,17,22,23]. Specifically, a representative image is selected from a large amount of unlabeled data and annotated by an oracle (called the annotator), thus becoming the labeled data. There are multiple approaches to using pool-based sampling.

Some studies focus on how to map the distance of a distribution to the information content of data points [24,25,26], while others estimate the diversity of a distribution by observing the gradient [27], future errors [28], or changes in the output of the trained model [29].

There are proposed active learning methods [22,23,30,31,32] with a pool-based sampling approach. However, these methods are computationally inefficient for current deep networks and large datasets.

To solve this problem, active learning methods [12,13,14,15,16,17] effective for deep networks and large datasets have appeared. Ref. [17] proposed the learning-loss method. Their model consists of a task module and a loss prediction module that predicts the loss of the task module. These two modules are learned simultaneously, and the loss prediction module is trained to estimate the target loss for unlabeled samples based on mid-layer feature information as a proxy for model uncertainty. Since the accuracy of loss prediction is affected by the performance of the task module, there is a disadvantage that if the task module is inaccurate, the predicted loss will not reflect how informative the sample is.

The methods that have become mainstream in recent years are those that use adversarial training [12,13,14,15,16]. In these methods, a variational autoencoder (VAE) reduces an image to a low-dimensional representation, and the discriminator determines whether the obtained representation was derived from a labeled or unlabeled image. By training the VAE and discriminator adversarially, we can select images from the unlabeled data pool that have features that the model has not yet seen. Because these methods sample images based on image features, they not only have the advantage of performing task-independent active learning, such as classification and segmentation, but also the disadvantage of taking more time than task learning. SRAAL [13] is an improved method of VAAL [12]. By relabeling the output of the discriminator with an online uncertainty indicator (OUI), SRAAL achieves better selection of uncertain images than VAAL. We think that the contribution of SRAAL is more in OUI than in VAE. OUI is used to relabel the discriminator, which is the same as distilling the discriminator at the output of OUI. In other words, the contribution that represents uncertainty can be found in the OUI. The verification of this hypothesis is one of the significant aspectsof our research. TA-VAAL [15] is a state-of-the-art method that combines VAAL [12] and learning-loss [17]. By changing the task-learning-based loss prediction to ranking loss prediction and embedding normalized ranking loss information into VAAL using a ranking-conditional generative adversarial network, the data distribution of both labeled and unlabeled pools is taken into account. There are also methods that use VAE [33,34] or GAN [35,36] to generate new images that are more informative for the current model. However, these synthesis methods have disadvantages such as high computational complexity and unstable performance [37].

These studies rely on specialized learning methods such as adversarial learning. This tends to result in complex and inefficient pipelines that are difficult to use in practical applications. Specifically, they have a model for computing uncertainty in addition to the task model, and since these are independent of each other, it is difficult for the task model to select the images it really needs, and the number of computation nodes is large, which increases the computation time. The disadvantage of the time required is that it detracts from the objective of improving the efficiency of annotation.

To solve this problem, Refs. [18,19,20] have been proposed to compute uncertainty algorithmically, using only task model learning. These methods do not require learning to predict uncertainty, making them easy to use in practice. We propose an uncertainty indicator generator (UIG), which is an algorithmic method similar to these methods.

The uncertainty sampling approach [38,39] annotates the data with the highest level of uncertainty in the results when inferring the model. Essentially, uncertainty is the degree to which the model cannot recognize the image. In other words, a sample that is closer to the decision boundary of the model can be regarded as more uncertain [40,41]. Therefore, our study focused on selecting samples around the decision boundary.

## 3. Proposed Method

### 3.1. Overview of the Proposed Method

As shown in Figure 3, our method is based on active learning with pool-based uncertainty sampling. This involves classification, multi-label classification, and semantic segmentation tasks. In Figure 3, the red arrows show the path for the unlabeled images, and the blue arrows show the path for the labeled images. First, we trained a task model using a small number of images from the labeled set. Images from the unlabeled set are then used as input in the uncertainty indicator generator (UIG) using the trained task model, with the uncertainty indicator for each image as output. The top K uncertain indicators are selected, annotated, and added to the labeled pool. By repeating this process, high performance is obtained with minimal annotation costs. The sampling flow using UIG is also shown in Algorithm 1.
**Algorithm 1** Sampling from Uncertainty Indicator**Input:** DL: labeled data pool, DU: unlabeled pool, *T*: random initialized task model, *C*: number of cycles in active learning, *N*: number of samplings;**Output:** Final learned parameters of *T*;
1:**for** i = 1 to *C* **do**2:   train *T* with DL3:   **for** xi in DU **do**4:     V = T(xi)5:     indicatori = UIG(V)6:     **if** (indicatori is top-N) **then**7:        yi = ORACLE(xi)8:        DL = DL + (xi,yi)9:        DU = DU − xi10:     **end if**11:   **end for**12:**end for**

### 3.2. Uncertainty Indicator Generator

We defined UIG based on the hypothesis that the contribution in SRAAL [13] is more in OUI than in VAE. In SRAAL, OUI is used to relabel the discriminator, which is the same as distilling the discriminator at the output of OUI. In other words, the contribution that represents uncertainty can be found in the OUI.

UIG can be used for multi-label classification in addition to the usual one-class classification task. The fundamental idea behind both UIG is the same: to compute uncertainty by using least confident and margin sampling. Least confident is a method of selecting images in which the maximum value of V is the smallest, and margin sampling is a method of selecting images in which the difference between the maximum value of V and the second largest value of V is the smallest. Note that these methods are usually used for classification tasks, and the proposed method is not the same as them in that it can be used in multi-label classification and uses the entire probability vector to compute the uncertainty indicator. Our method can be described as a modified version of the least confident and margin sampling methods. It is designed to express the uncertainty of each image in the unlabeled pool with a specific importance using continuous values in the range [0, 1). The UIG calculates the uncertainty indicators based on the prediction vectors of the task model (image classifier and semantic segmentation model). Specifically, in the case of image classification, the prediction is a probability vector for each category. In segmentation, each pixel has a probability vector, and the prediction vector is the average value of each probability vector. The calculation of the uncertainty indicator can be formulated as
(1)$$Indicator\left(x_{U}\right)\;=\;1\;-\;\frac{MINVar(V)}{Var(V)}\;\;\times \;\prod_{i=1}^{m}max_{i}\left(V\right),$$
where $x_{U}$ is the unlabeled image, and $V\;=\;p\left(x_{U}|D_{L}\right)$ is the probability vector of $x_{U}$ based on the trained task model in the current labeled pool $D_{L}$. $max_{i}\left(V\right)$ is the *i*-th largest value of *V*, and *m* is the number of classes chosen from that image. For general classification tasks and semantic segmentation, it is 1. For multi-class classification, m takes different values for each image and is determined by the equation as $max_{m}\left(V\right)\;\geq \;0.5\;>\;max_{m+1}\left(V\right)$. Assuming that the number of classes is *C*, MINVar(V) can be formulated as
(2)$$MINVar\left(V\right)\;=\;Var\left(V^{\prime }\right)\;=\;\frac{1}{C}\left(\sum_{i=1}^{m}{\left(\frac{sum(V)}{C}\;-\;max_{i}\left(V\right)\right)}^{2}\;+\;\left(C\;-\;m\right){\left(\frac{C\sum_{i=1}^{m}max_{i}\left(V\right)\;-\;m\;\times \;sum\left(V\right)}{C\;-\;m}\right)}^{2}\right)$$

MINVar(V) is the variance of the vector $V^{\prime }$, where the top *m* elements are the same as *V*, and the other C − m elements are $\left(\sum_{i=m+1}^{i=C}{max}_{i}\left(V\right)\right)/\left(C\;-\;m\right)$.

$V^{\prime }$ is divided by the mean of the possible answer (top-m) and the rest, so

MINVar(V)/Var(V) can be considered as margin sampling for the top-m and the rest of the whole. The more the value of VAR(V) differs from MINVAR(V), the higher the uncertainty is considered. MINVar(V)/Var(V) indicates the degree of concentration for the probability distribution. $\prod_{i=1}^{m}max_{i}\left(V\right)$ is considered more uncertain for smaller values of top-m. This part indicates least confidence. From the above, we can see that the uncertainty indicator has the following three characteristics, and a boundary of [0, 1), a negative correlation with the value of the maximum probability, and a negative correlation with the degree of concentration for the probability distribution. In other words, the image is measured in terms of uncertainty from the two perspectives of least confident and margin sampling. The closer the uncertainty indicator is to one, the more uncertain the image is, and the more likely it is to be selected. As a concrete example, take the three-class classification of dogs, cats, and tigers. Consider that the output of the task model for a certain image is [0.6,0.1,0.3]. In this case, V = [0.6,0.1,0.3], and sincem = 1, $V^{\prime }is\left[0.6,0.2,0.2\right]$. Since VAR is the variance of *V* and MINVAR is the variance of $V^{\prime }$, they are 0.042 and 0.036, respectively. Additionally, $max_{1}\left(V\right)$ is the maximum value of *V*, which is 0.6. Applying these values to Equation (Equation 1), we obtain an uncertainty indicator of 0.49. If this is calculated for an image such as V = [0.4,0.4,0.2], where it is not known whether the image is of a dog or a cat, the indicator is obtained as 0.90, which can be regarded as high uncertainty.

## 4. Experiment

### 4.1. Experimental Overview

In this study, we evaluate the proposed method in three tasks: classification, multi-label classification, and semantic segmentation, and compare the results with those obtained by sampling with four different methods: VAAL [12], SRAAL [13], TA-VAAL [15], entropy [6], and random. The results are compared and evaluated. Furthermore, to confirm that the proposed method is able to select more uncertain images than the conventional method, we compare the uncertainty indicator with the real loss. We also measure the execution time for each dataset and compare it with VAAL and SRAAL.

We also confirm in our experiments if we are able to select samples near the boundaries and if we are able to select samples with higher actual losses.

In addition, since UIG is a combination of least confident and margin sampling, we performed ablation studies for each of them. Specifically, the experiment was conducted with $Indicator\left(x_{U}\right)\;=\;1\;-\;max\left(V\right)$ for least confident and $Indicator\left(x_{U}\right)\;=\;1\;-\;MINVar\left(V\right)/Var\left(V\right)$ for margin sampling.

We started the experiment with 10% of the total dataset randomly sampled as labeled data and the remaining 90% as unlabeled data. We continued sampling in 5% increments until the labeled data reached 40% of the total dataset. Afterwards, we compared the performance of each dataset with that of previous studies and random sampling. Because active learning aims to achieve high accuracy with even a small number of annotations, in this experiment, as in [12,13], we limited the labeled data to 40% of the total data. For multi-label classification, we start with 1% of the labeled data and sample in 0.5% increments until we reach 4%, and for semantic segmentation, we start with 10% of the labeled data and sample in 10% increments until we reach 70%.

The experiments were each conducted five times, and the results of the experiments were averaged to determine the final value. The evaluation metrics used were accuracy for the classification tasks and mean intersection over union (mIOU) for the semantic segmentation task. Since the purpose of active learning is to improve the efficiency of annotation in normal learning, commonly used metrics were selected for each evaluation metric.

### 4.2. Dataset

In Table 1, we show configuration of the datasets we used. For the image classification task in this study, we used the CIFAR-10 [42] and CIFAR-100 [42] datasets. In the semantic segmentation task, we used the Cityscapes [43] dataset. Lastly, for the multi-label classification task, we used the CelebA [44] dataset. CIFAR-10 and CIFAR-100 each contained 60,000 32 × 32 × 3 images. A total of 50,000 were training data, and 10,000 were test data. As indicated in their names, CIFAR-10 had 10 classes, and CIFAR-100 had 100 classes. When training the model, Cityscapes converted the image sizes to 688 × 688 × 3. A total of 2975 images were used as training data and 500 images were used as test data. Cityscapes was annotated with 34 classes, including ambiguous classes. For this study, the model was converted to 19 non-ambiguous classes. Upon training the model, Celeb A converted the image size to 64 × 64 × 3 with 40 classes and only few labels in these classes used as answers. There were 162,770 images in this dataset, making it much larger than the others. Therefore, for this dataset, the sampling rate was changed 0.5% from 1% to 4% in this study. To confirm if we were able to select samples near the boundaries, we used the Scikit-learn makemoons library [21] to generate a two-dimensional binary dataset for binary classification as shown in Figure 2. The noise option was set to 0.2 and the size of the labeled and unlabeled datasets was set to 500 samples each. From the 500 unlabeled samples, 30 samples were selected for each method.

### 4.3. Implementation Details

For each task, we trained about 100 epochs using the training data of each dataset. The task models used for training were VGG16 [45] for the image classification task and the multi-label classification task. We also used a dilated residual network (DRN) [46] for the semantic segmentation task.

The optimization algorithms were stochastic gradient descent (SGD) for the image classification task and the multi-label classification task, and Adam was used for the semantic segmentation task, with learning rates of 0.01 and 0.0003, respectively. We used cross-entropy error for the image classification task and the semantic segmentation task, and negative log likelihood (NLL) losses were used for the multi-label classification task as the loss functions. In addition, we use only RandomHorizontalFlip as data augmentation.

Since the experiments in Section 5.5 use a simplified dataset, a simplified model was also used. The task model used a three-layer multilayer perceptron (MLP) with a learning rate of 0.1 and the SGD optimizer. SRAAL’s VAE used a two-layer MLP with ReLU for the encoder and decoder, respectively, and a two-layer MLP for the discriminator. The VAE and discriminator used the Adam optimizer. All epochs were set to 100 for the experiments.

Note that UIG does not require hyperparameters.

## 5. Results

### 5.1. Experimental Results for CIFAR-10

The experimental results with CIFAR-10 are shown in Figure 4a. When we trained all the data in CIFAR-10 with the VGG16, the accuracy was 84.50%. The proposed method achieved 81.83% with 40% of the data using the same task model. When we randomly sampled 40% of the data, the accuracy was 76.69%. This was almost equal to the accuracy of the proposed method when 25% of the data was sampled. The accuracy of the conventional method, VAAL [12], was 78.96% with 40% of the data, which was close to the proposed method’s accuracy of 30%.

The accuracy of SRAAL is higher than that of VAAL, but lower than that of the proposed method, which has a more simpler structure. This is because SRAAL structurally trains the model using the output of the UIG as the correct label, and as a result, the uncertainty can be expressed more accurately by using the UIG alone. The accuracy of TA-VAAL is higher than that of SRAAL, but lower than that of the proposed method. It should be noted that the results are very close to the proposed method at 15%.

Figure 5a shows the improvement from the random sampling baseline for each method with standard deviation (shaded). The proposed method outperformed others on CIFAR-10 in all stages.

These results indicate that in 10-class classification, selecting samples near the decision boundary of the model is very effective in learning the model. Based on these results, we can say that our method is able to reduce the amount of samples to be annotated, which is the objective of active learning.

The accuracy of each method differs from that of the original paper due to the different models used in the task model (ResNet was used in SRAAL, while VGG was used in this experiment). In particular, SRAAL does not include a description of data augmentation, which suggests that the experimental environment other than the method is different. Because we conducted our experiments in the same environment for each method, the results may differ from those in the original paper.

### 5.2. Experimental Results for CIFAR-100

The experimental results with CIFAR-100 are shown in Figure 4b and Figure 5b. When the VGG16 was trained on all the data in CIFAR-100, the accuracy was 51.59%. The proposed method achieved 39.66% with 40% of the data using the same task model. When we randomly sampled 40% of the data, the accuracy was 36.52%. This was almost equal to the accuracy of the proposed method when sampling 35% of the data. In VAAL, the accuracy was 37.43% for 40% of the data, which was also close to the proposed method’s 35%.

These results show that the accuracy of the proposed method was better than the other methods except TA-VAAL, even with more complex datasets that had more classes and a similar amount of data to CIFAR-10. The results also indicate that, compared to CIFAR-10, the proposed method was effective even before the convergence of the accuracy values.

For sampling after 10%, the proposed method showed better accuracy than the previous studies. This is because the proposed method computes uncertainty using the output of the task model, and thus provides information lacking in the task model, while VAAL and SRAAL do not necessarily contribute to learning the task model, as they provide a wide variety of data by VAE, which is not the task model. The accuracy of TA-VAAL is higher than that of the proposed method at 10%, 15%, and 40%. The values of 10% and 40% are almost the same, but the value of 15% is remarkable. Accuracy was also higher in CIFAR-10 when sampling at 15% than at other sampling times. In combination with the accuracy of the original paper, TA-VAAL seems to be more effective when the number of labeled sets is small. In contrast, the proposed method is stable and highly accurate at all sampling points.

On CIFAR-100, TA-VAAL showed comparable performance to the proposed method. However, TA-VAAL takes about 10 times longer than the proposed method for training, making it computationally expensive.

### 5.3. Experimental Results for Cityscapes

The experimental results with Cityscapes are shown in Figure 4c and Figure 5c. When a DRN was trained on all the data in Cityscapes, the accuracy was 58.14%. The proposed method achieved 56.72% with 70% of the data using the same task model. When we randomly sampled 70% of the data, the accuracy was 54.98%. This was almost equal to the accuracy of the proposed method when sampling 30% of the data. The conventional method, VAAL, achieved 56.10% accuracy with 70% of the data, which was close to the proposed method’s accuracy of 50%. These results show that the proposed method performed better than the conventional method in semantic segmentation.

The Cityscapes results in Figure 5c appear to decrease in accuracy as sampling progresses compared to the results for the other datasets. The Cityscapes dataset sampled up to 70% is more similar to random sampling than the other datasets.

Table 2 shows a comparison of the average IOUs for each class obtained by random sampling and the proposed method; each sampling was performed five times with 50% sampling. Table 3 shows the average, minimum, and maximum values of IOUs for each class in random sampling.

From the results shown in Table 2, the proposed method obtained higher IOUs than the random sampling in 15 of the 17 classes (the two exceptions were “Road” and “Fence”). In addition, the mean of the maximum value of each class in Table 3 was 53.07, while the mean of the proposed method was 52.90. This was close to the mean of the random sampling, indicating that the proposed method had ideal sampling.

Figure 6 compares the ground truth images and the training results using all training images in Cityscapes and the training results using 50% sampling by the proposed method. The proposed method can almost obtain the same results as those obtained by training with all the data, but only using half of the data.

These quantitative and qualitative comparisons show the superiority of the proposed method.

The average values in Table 2 differ from those in Figure 4c at 50% sampling because the values in Figure 4c are the mIOUs of each sample averaged over all the samples. The value for each class in Table 2 is the average of the IOUs of all the samples for each class, and the AVERAGE values are the average of each class (accounting for the different number of occurrences).

### 5.4. Experimental Results for CelebA

The results of the CelebA experiment are shown in Figure 4d and Figure 5d. When the VGG16 was trained on all the data of CelebA, the accuracy was 88.40%. The proposed method achieved 87.71% with 4% of the data using the same task model. When 4% of the data was sampled randomly, the accuracy was 87.36%. The conventional method, VAAL, achieved 87.42% accuracy with 4% of the data.

After 2.5%, there was little difference between VAAL and random sampling, with the proposed method achieving higher accuracy on a relatively stable basis. These results indicate that the proposed method outperforms the conventional methods in multi-label classification.

The accuracy of SRAAL is lower than that of other methods, which indicates that the uncertainty indicator inferred by SRAAL does not work well for multi-label classification.

In addition, there seemed to be variations in the results of the proposed method, VAAL, and random sampling with this dataset that was not seen in the other datasets. In the case of CelebA, the accuracy was close to convergence at 1% sampling. The difference in performance between the proposed method and the other methods was smaller with this dataset than with the other datasets. However, the advantage of the proposed method is the ability to obtain higher accuracy than the other methods on average, even if the range of increase was small.

### 5.5. Analysis of Selected Samples

Figure 2 shows which samples were selected in the two-dimensional dataset for SRAAL and the proposed method. The samples selected for SRAAL are spread over the whole area, whereas the proposed method selects samples close to the decision boundary line. Our method aims at efficient annotation by selecting samples near the decision boundary, and Figure 2 shows that our method actually selects samples closer to the decision boundary than conventional methods.

### 5.6. Analysis of Uncertainty Indicator

Figure 7 shows a graph of the actual losses of SRAAL and the proposed method on the *x*-axis and the uncertainty index of each method on the *y*-axis in CIFAR-10. The point chosen for the 5% selection after training with the randomly chosen 10% is shown in red. Note that both SRAAL and the proposed method select data points with the highest uncertainty index for unlabeled data. In this figure, it is ideal to be able to sample the points on the right (where the real losses are higher). SRAAL selects images with a wide range of actual losses, as shown in Figure 7a, whereas our proposed method can select images with relatively high real loss values, as shown in Figure 7b. Furthermore, the correlation coefficient of the proposed method is 0.43, compared to 0.01 in SRAAL. From the above, it appears that the proposed method performs better than SRAAL on a variety of tasks.

### 5.7. Comparison of Model Computational Complexity

We compared the execution time of the proposed method with that of VAAL and SRAAL, and the results are shown in Table 4. The experimental results are measured on an NVIDIA GeForce RTX 2080Ti. We succeeded in reducing the execution time for each dataset approximately by the following factors: 10 for both CIFAR-10 and CIFAR-100, two for Cityscapes, and five for CelebA. The memory requirements for sampling were 4114 MiB for the proposed method, and 5586 MiB for VAAL and SRAAL when mini-batch size was 128.

### 5.8. Ablation Study

In order to evaluate the contribution of least confident and margin sampling in the proposed method (UIG), we compared least confident, margin sampling, and full UIG using CIFAR-10. Specifically, the experiment was conducted with $Indicator\left(x_{U}\right)\;=\;1\;-\;max\left(V\right)$ for least confident and $Indicator\left(x_{U}\right)\;=\;1\;-\;MINVar\left(V\right)/Var\left(V\right)\;$ for margin sampling. The results are shown in Table 5. The full UIG was the highest for sampling after 10%, indicating the effectiveness of both least confident and margin sampling.

## 6. Limitation

Recent active learning methods require a great deal of time to learn a model, and it would be faster to perform annotation in a straightforward manner. We proposed a method that requires no learning other than the task model and solved the problem. We believe that the proposed method is the best method for active learning. However, compared to other, small-data learning methods, the improvement in accuracy possible with active learning is small. The proposed method is no exception. Recently, semi-supervised learning [4,5] and unsupervised learning [1,2,3] have been showing good results. While active learning selects valid images from unlabeled images, these methods use all images for learning. Active learning is suitable for physically expensive use cases such as semantic segmentation, while semi-supervised or unsupervised learning is often more suitable for cases that require expert knowledge, such as medical image diagnosis. We need to consider useful methods for combining active learning with these methods for real-world use.

## 7. Conclusions

In order to select the most informative sample from an unlabeled pool, we proposed a model to derive the most informative unlabeled sample from the output of the task model, a UIG. Experimental results show that the proposed method is able to select images with higher loss than the conventional method, and the annotation cost can be reduced by minimizing the amount of labeled data that is used. The proposed method outperforms the current SoTA method by 1% accuracy in CIFAR-10. We also succeeded in reducing the execution time of the models by about 90% for CIFAR-10 and CIFAR-100, about 50% for Cityscapes, and about 80% for CelebA, as compared to the conventional methods.

In this study, we have shown the superiority of our method for images, but we believe that it can also be applied to more advanced tasks, such as video. We will also consider useful methods for combining active learning with semi-supervised and unsupervised learning for real-world applications.

## Figures and Tables

**Figure 1 sensors-22-05244-f001:**
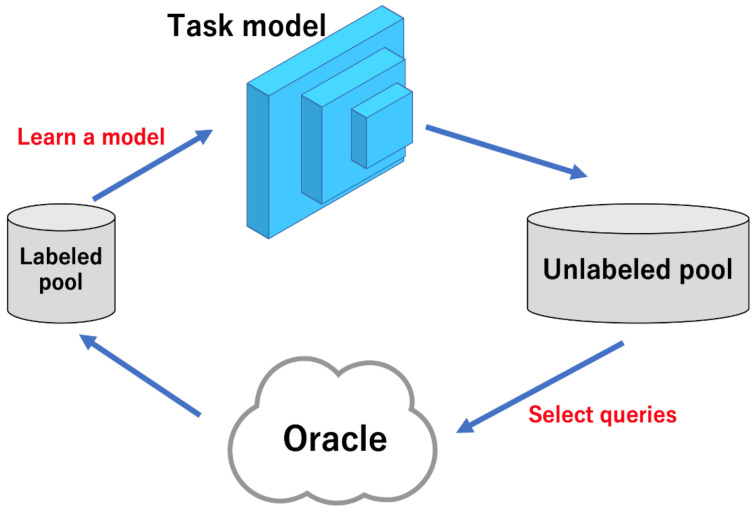
Traditional pool-based active learning cycle. In each iteration, a task model is trained using the labeled images. After training, unlabeled images are selected based on the model’s inference and then labeled by the oracle. The active learning scheme performs this iteratively until the model performance meets the user’s requirements or the label budget runs out.

**Figure 2 sensors-22-05244-f002:**
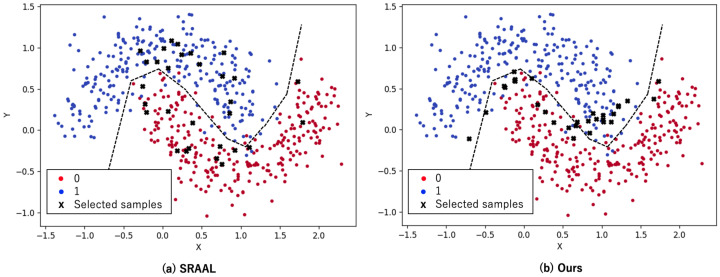
Visual results of active learning methods (SRAAL [13], our UIG) on a toy example. The red points represent classes with 0, and blue points with 1. The black faction marks are the samples selected by each method. The oracle decision boundary is shown as a black dashed line. We trained on 500 samples and selected 30 samples from the 500 unlabeled samples. It can be seen that the proposed method is able to select samples near the decision boundary better than SRAAL. Reprinted/adapted with permission from Ref. [21]. 2007–2021, the scikit-learn developers.

**Figure 3 sensors-22-05244-f003:**
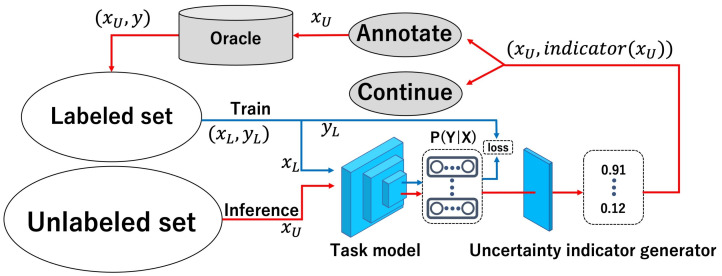
Model of our proposed method. The red arrows show the path for the unlabeled images, and the blue arrows show the path for the labeled images. The labeled images are used to train the task model. Using the output of the trained task model, the uncertainty indicator generator (UIG) outputs the uncertainty indicator for unlabeled images. Unlabeled images with high uncertainty are labeled first.

**Figure 4 sensors-22-05244-f004:**
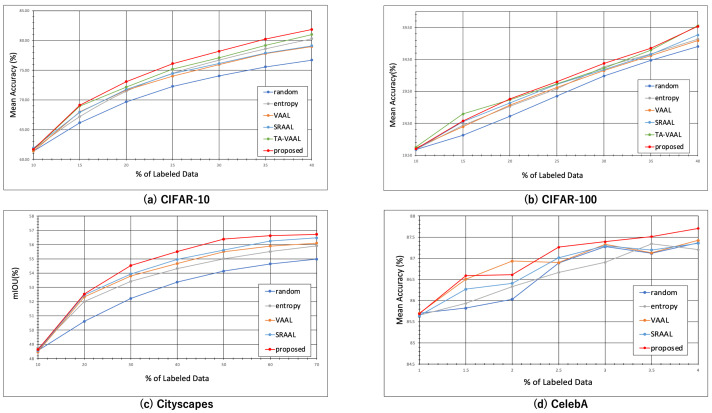
Results of experiments. We show the results for each dataset, starting from 10% randomly sampled and then 5% sampled by each method. Our method outperformed others on each dataset in many stages.

**Figure 5 sensors-22-05244-f005:**
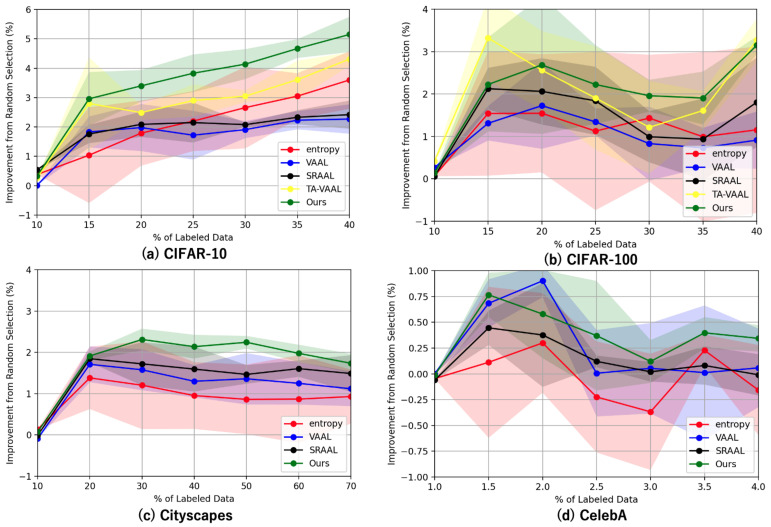
Results of experiments. Mean accuracy improvements with standard deviation (shaded) of active learning methods from random sampling baseline over the number of labeled samples. Differences in accuracy between methods are important for comparisons, so they are featured.

**Figure 6 sensors-22-05244-f006:**
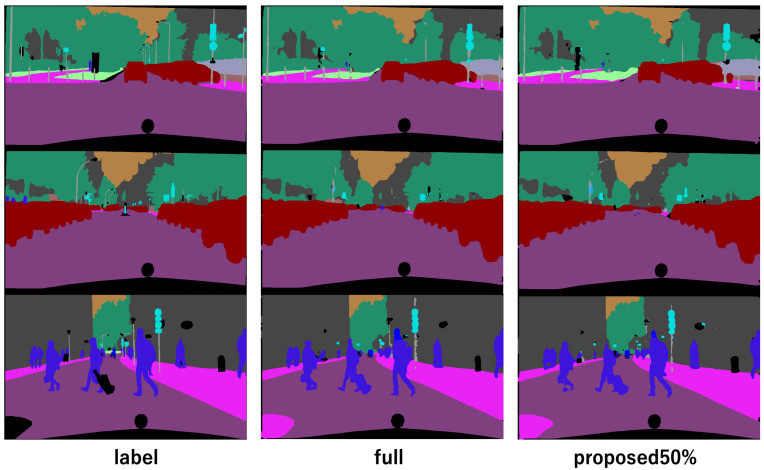
Qualitative evaluation of Cityscapes. Left is ground truth, center is result of all training images, right is result of 50% sampling by proposed method. Using the proposed method, we can obtain a semantic segmentation image with half the annotations, which is almost the same as when training using all images.

**Figure 7 sensors-22-05244-f007:**
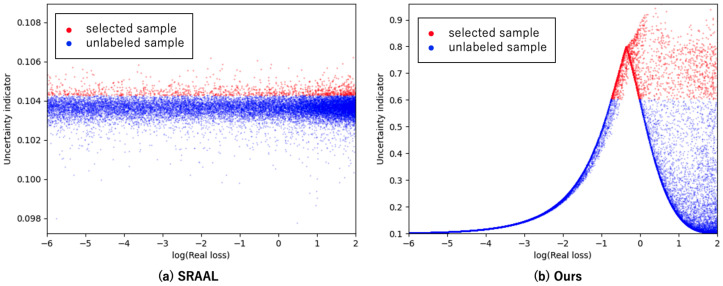
Relationships between real loss (task model uncertainty) and uncertainty indicator in (**a**) SRAAL and (**b**) our UIG. The red point should be in the upper right corner of the figure.

**Table 1 sensors-22-05244-t001:** Configuration of the datasets. It shows the task used and the sampling interval for each dataset.

Dataset	Task	Sampling Interval
CIFAR-10 [42]	Classification	5% (10% to 40%)
CIFAR-100 [42]	Classification	5% (10% to 40%)
Cityscapes [43]	Semantic segmentation	10% (10% to 70%)
CelebA [44]	Multi-label classification	0.5% (1% to 4%)

**Table 2 sensors-22-05244-t002:** IOUs per class. Comparison of results for random sampling and the proposed method. The higher values are bolded.

Class	Random 50%	Proposed 50%	Occurrence
Road	**93.75**	92.04	96
Sidewalk	65.69	**67.90**	92
Building	79.06	**83.82**	98
Wall	22.83	**27.68**	39
Fence	**27.52**	27.26	37
Poll	27.77	**32.14**	99
Traffic light	32.33	**46.37**	60
Traffic sign	45.89	**52.87**	96
Vegetation	84.46	**88.15**	96
Terrain	30.00	**32.89**	47
Sky	85.26	**89.28**	91
Person	40.18	**55.92**	82
Rider	21.25	**28.84**	48
Car	80.70	**83.53**	97
Truck	19.67	**19.96**	21
Bus	29.21	**46.36**	17
Train	42.01	**46.61**	2
Motorcycle	20.90	**30.76**	11
Bicycle	42.82	**52.77**	65
Average	46.91	**52.90**	-

**Table 3 sensors-22-05244-t003:** IOUs at 50% random sampling. Mean, minimum, and maximum values for five random sampling runs. This table shows how much mIOU can be taken in each class of random sampling.

Class	Average	Min	Max
Road	93.75	91.36	94.66
Sidewalk	65.69	64.56	66.60
Building	79.06	77.53	83.58
Wall	22.83	19.91	28.11
Fence	27.52	26.32	29.16
Poll	27.77	25.86	30.83
Traffic light	32.33	28.64	42.76
Traffic sign	45.89	43.39	50.16
Vegetation	84.46	83.10	88.00
Terrain	30.00	27.87	35.84
Sky	85.26	82.92	90.44
Person	40.18	35.29	54.52
Rider	21.25	14.25	25.59
Car	80.70	79.06	84.42
Truck	19.67	11.13	27.78
Bus	29.21	25.29	34.78
Train	42.01	31.97	62.79
Motorcycle	20.90	16.39	25.23
Bicycle	42.82	38.56	53.07
Average	46.91	43.34	53.07

**Table 4 sensors-22-05244-t004:** Comparison of execution time. This shows that the proposed method is able to execute the sampling pipeline faster than the conventional method. The faster values are bolded.

Dataset	VAAL (s)	SRAAL (s)	Proposed (s)
CIFAR-10	$1.9\;\times \;10^{5}$	$2.2\;\times \;10^{5}$	$2.0\;\times \;10^{4}$
CIFAR-100	$2.8\;\times \;10^{5}$	$2.9\;\times \;10^{5}$	$2.2\;\times \;10^{4}$
Cityscapes	$3.0\;\times \;10^{5}$	$7.0\;\times \;10^{5}$	$1.3\;\times \;10^{5}$
CelebA	$4.0\;\times \;10^{5}$	$1.2\;\times \;10^{6}$	$8.3\;\times \;10^{4}$

**Table 5 sensors-22-05244-t005:** Experiment result for ablation study on CIFAR-10. It shows the contribution of least confident and margin sampling in the proposed method. The higher values are bolded.

Data	Least Confident	Margin Sampling	Full
10%	61.70	**61.73**	61.69
15%	68.65	68.68	**69.12**
20%	72.55	72.74	**73.06**
25%	75.73	76.01	**76.10**
30%	77.81	77.77	**78.18**
35%	80.20	80.05	**80.22**
40%	81.59	81.65	**81.83**

## Data Availability

Data available in publicly accessible repositories. The data presented in this study are openly available in Refs. [42,43,44].

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
