# Peer review of "Non-Deep Active Learning for Deep Neural Networks"

_sensors, 2022, doi:10.3390/s22145244_

Round 1

Reviewer 1 Report

The research focuses on the non-active learning for deep neural networks. There are some areas of improvement, which are given as under:

1. The introduction is inadequate and needs improvement. The gaps in the state of research at the moment are not entirely clear. The authors should make an attempt to address it in the introduction.

2. There are terms used in introduction like supervised learning, unsupervised learning etc, but not defined and explained. Similarly, terms like classification, segmentation etc. needs to be defined and explained.

3. The introduction is short and needs enrichment in content. The significance of non-deep learning needs to be developed on a stronger footing and related literature needs to be cited.

4. Authors need to enhance the related works and summarize it using a table, so that the prior research and how this research would take it forward becomes clear. The authors need to cite the relevant works more, particulary in the last 3 to 5 years.

5. Please provide source for Figure 1.

6. The study's implications should be discussed in greater detail in the conclusion.

Reviewer 2 Report

In this active learning paper, the authors propose to use a uncertainty indicator to identify high content images which needs to be labelled for best learning. The authors show comparison with some other works however my concern is regarding the claim of improving SOTA.

Authors claim to improve the SOTA by 1%  however following works report higher accuracy with similar amount of data:

a) Beluch, William H., et al. "The power of ensembles for active learning in image classification." Proceedings of the IEEE conference on computer vision and pattern recognition. 2018.

b) Guo, Jifeng, et al. "Redundancy Removal Adversarial Active Learning Based on Norm Online Uncertainty Indicator." Computational Intelligence and Neuroscience 2021 (2021).

Also, could you please provide some examples where authors have used/proposed additional deep networks to identify the samples worth annotating.

Reviewer 3 Report

Some major points are given below:

 1. The objective of the work should be highlighted in the introduction section.

2. For the developed method, do you consider the computation burden issue? For this case, i think you need to discuss more after the main results and a remark will be helpful here. What the difficulties you have met when deriving the current results? The authors are suggested to add a remark after the main results.

3. In the experimental part, the detail parameters used in the proposed methodology are not given.

4. The authors have hardly taken effort to discuss the proposed work. This demotivates the reader. It should be more clearly discussed.

5.  The results are not clearly explained in alignment to the objective of the paper.

6.  Some standard metrics are used by the authors to measure the performance of the proposed method. What are the standard acceptable values of performance metrics? Why authors choose performance metrics? Must explain the reasons.

7. Abbreviations throughout text need to be revised. In some instances, abbreviations are mentioned without the original phrase.

8. The computational complexity either in terms of calculations required or in terms of execution time may also be computed and compared.

Minor Comments:

*In Section 1, it is better Section Contributions follows Section Related Works. The authors did not clearly investigate the relate works. It is better the authors investigate all the related works and then state their contributions.

* The contributions and the hypothesis should be presented clearly in a concise way,

* To have an unbiased view in the manuscript, there should be some discussions on the limitations of the proposed method. Additionally, clearly justify the novelty and innovative insights of this manuscript,

* Revisions in terms of detailed experiments and discussions can be re-checked,

* Discussions of the experimental results achieved can be enhanced, being objective and concise,

* The innovation of the strategy/method is not clear described enough, need to provide more details and explain further,

* The linguistic quality needs improvement. It is essential to make sure that the manuscript reads smoothly- this definitely helps the reader fully appreciate your research findings. 

* Add industrial significance of the proposed approach,

Reviewer 4 Report

The manuscript sounds technically poor, I have following concerns should be addressed before any decision.  The paper currently need revision.

*The existing literature should be classified and systematically reviewed, instead of being independently introduced one-by-one.

*The abstract is too general and not prepared objectively. It should briefly highlight the paper's novelty as what is the main problem, how has it been resolved and where the novelty lies?

*The 'conclusions' are a key component of the paper. It should complement the 'abstract' and normally used by experts to value the paper's engineering content. In general, it should sum up the most important outcomes of the paper. It should simply provide critical facts and figures achieved in this paper for supporting the claims.

*For better readability, the authors may expand the abbreviations at every first occurrence.

*The author should provide only relevant information related to this paper and reserve more space for the proposed framework.

*However, the author should compare the proposed algorithm with other recent works or provide a discussion. Otherwise, it's hard for the reader to identify the novelty and contribution of this work.

*The descriptions given in this proposed scheme are not sufficient that this manuscript only adopted a variety of existing methods to complete the experiment where there are no strong hypothesis and methodical theoretical arguments. Therefore, the reviewer considers that this paper needs more works.

*Key contribution and novelty has not been detailed in manuscript. Please include it in the introduction section

*What are the limitations of the related works

*Are there any limitations of this carried out study?

*How to select and optimize the user-defined parameters in the proposed model?

*There are quite a few abbreviations are used in the manuscript. It is suggested to use a table to host all the frequently used abbreviations with their descriptions to improve the readability

*Explain the evaluation metrics and justify why those evaluation metrics are used?

*Some sentences are too long to follow, it is suggested that to break them down into short but meaningful ones to make the manuscript readable.

*The title is pretty deceptive and does not address the problem completely.

*Every time a method/formula is used for something, it needs to be justified by either (a) prior work showing the superiority of this method, or (b) by your experiments showing its advantage over prior work methods - comparison is needed, or (c) formal proof of optimality. Please consider more prior works.

*The data is not described. Proper data description should contain the number of data items, number of parameters, distribution analysis of parameters, and of the target parameter itself for classification.

*The related works section is very short and no benefits from it. I suggest increasing the number of studies and add a new discussion there to show the advantage

*Use Anova test to record the significant difference between performance of the proposed and existing methods.

Round 2

Reviewer 2 Report

I understand that data labeling is time consuming and we need better approaches of active learning to help identify/select data. On my previous question, the authors state the difference of models. Testing the performance of your approach on CIFAR -10 with other models should not be tedious as you already selected the samples. I have few more comments/questions:

1. Uncertainty sampling approaches as also mentioned in the paper  are not novel [35][36]. Authors propose a UIG and claim that as novelty of this works. The least confident and margin sampling is not novel thus I am concerned about the novelty of this work. Please explain and if needed improve the write-up.

2. Please compare your results with similar works that use uncertainty sampling.

3. Figure 6 shows that the sampling strategy as mentioned in the paper also matches with the loss values. Could you please provide a comparison baseline with loss value as selecting parameter for sample selection?

Reviewer 4 Report

The paper still needs revision.

11. Please explain in your captions of figure and title of table, why are these tables or figures necessary in your paper? What are the purposes and what are the message you want to deliver via these figures and tables?

22. The current metrics might not be sufficient to judge the performance of the model holistically. Please enhance the result analysis part of your paper.

43. What are the real-life use cases of the proposed model? The authors can add a theoretical discussion on the real-life usage of the proposed model.

Round 3

Reviewer 2 Report

The authors have addressed my concerns/comments.

Reviewer 4 Report

The paper is well revised and can be accepted.